# Masculinity norms and occupational role orientations in men treated for depression

**Reinhold Kilian**[1]*, **Annabel Müller-Stierlin**[1], **Felicitas Söhner**[1,2], **Petra Beschoner**[3], **Harald Gündel**[3], **Tobias Staiger**[1], **Maja Stiawa**[1], **Thomas Becker**[1], **Karel Frasch**[1,4], **Maria Panzirsch**[4], **Max Schmauß**[5], **Silvia Krumm**[1]

**1** Department of Psychiatry and Psychotherapy II, Bezirkskrankenhaus Günzburg, Ulm University, Günzburg, Germany, **2** Department of the History, Philosophy and Ethics of Medicine, Centre for Health and Society, Düsseldorf University, Düsseldorf, Germany, **3** Department of Psychosomatic Medicine and Psychotherapy, Ulm University, Ulm, Germany, **4** Fachklinik für Psychiatrie, Psychotherapie und Psychosomatik an der Donau-Ries Klinik, Bezirkskrankenhaus Donauwörth, Donauwörth, Germany, **5** Department of Psychiatry, Psychotherapy and Psychosomatics, Bezirkskrankenhaus Augsburg, Augsburg University, Augsburg, Germany

* reinhold.kilian@uni-ulm.de

## Abstract

### Purpose

A traditional male role orientation is considered to increase the risk of depression and preventing men from disclosing symptoms of mental illness and seeking professional help. Less is known about the variance of masculinity orientations in men already treated for depression and their role in the treatment process. In this study, patterns of masculinity norms and work role orientations will be identified among men treated for depression. Associations of these patterns with depressive symptoms, stigma and delay in professional help-seeking will be investigated.

### Methods

In a cross-sectional study, male role orientations (MRNS), work-related attitudes (AVEM), symptoms of mental disorders (PHQ), and attitudes related to stigma of mental illness (DSS) were assessed by standardized methods in a sample of 250 men treated for depression in general medical, psychiatric and psychotherapeutic services. Data were analyzed by means of latent profile analysis (LPA), by multinomial and linear regression models, and by path analysis.

### Results

The results of LPA revealed three latent classes of men treated for depression. Men assigned to class one reported a less traditional male role orientation, low professional ambitions and low coping capacities; men assigned to class two reported a traditional masculinity orientation, high professional ambitions but low coping capacities; men assigned to class three reported less traditional masculinity tended orientations, medium professional ambitions and high coping capacities. Men assigned to classes one and two to have more

**Data Availability Statement:** All relevant data are within the paper and its Supporting Information files.

**Funding:** The study was funded by the Deutsche Forschungsgemeinschaft (DFG) to RK (KI 792/3-1); SK (KR 3879/2-1), and HG (GU 805/2-1). The funder had no role in study design, data collection and analysis, decision to publish, or preparation of the manuscript.

**Competing interests:** The authors have declared that no competing interests exist.

stigmatizing attitudes, longer periods of untreated illness and more severe symptoms of mental disorders, in comparison to men assigned to class three.

## Conclusions

Overall, this study reveals that traditional masculinity norms and work-role orientations in men treated for depression are associated with a worse mental health status. Our study results also suggest that a slackening of traditional masculinity norms is associated with improved psychological well-being if it does not coincide with a complete distancing from professional ambitions and a lack of ability to cope with professional stress.

## Introduction

While the worldwide depression prevalence among women is about twice that in men, suicide rates among men are significantly higher than among women [1–3]. In order to explain this paradoxical relation several authors hypothesize that a majority of men affected by depression regard their symptoms as incompatible with their masculine self-image and that they are therefore reluctant to seek help which in turn increases the risk of suicide [4–9]. This hypothesis is supported by study results indicating that adherence to a traditional masculine role orientation in depressed men was found to be related to increased self-stigma, which in turn worked as a mediator toward reduced willingness to seek help [10–11]. However, there is also evidence that the associations of traditional masculinity with adverse mental health behavior vary according to the specific set of masculine norms [9]. While adherence to masculine norms of self-reliance, power over women, and sexual dominance was found to be negatively associated with mental health, conformity to masculine norm of primacy of work was not related to any mental health-related outcome. Also, sexist and homophobic male attitudes appear to be associated with greater risk of poor mental health [11–12]. Recently, a prospective study on depression among male college students revealed that endorsement of self-reliance, playboy attitude and violent masculine norms increased the risk for depression, while adherence to winning and power over women was associated with lower depressive symptoms [13].

Several authors have hypothesized that gender-specific occupational role expectations and other workplace factors interact with the mental health impact of masculine role orientations [14, 15]. However, current findings about the associations between occupational role expectations and mental health are inconclusive. While studies from several countries [15] found that men working in male-dominated occupations or industries have a higher risk of depression than men working in occupations with a lower percentage of male workforce, results from an Australian national study on male health [16] revealed no direct associations between male workforce and mental health. A detailed analysis shows that it is not the male-dominated work environment per se but particular masculine attitudes among male workers that were associated with negative effects on their mental health [16]. While a high priority attributed to work was related to better mental health, other work-related attitudes such as self-reliance, dominance over others and winning had strong negative effects [16].

In sum recent research supports the hypothesis that adherence towards a traditional image of masculinity is related to lower male well-being and mental health [9, 11] and that this association is increased in male dominated occupational environments [15, 16]. However, in their consideration of what this implies for the development and the evaluation of treatment strategies, Seidler et al. [11] have criticized a one-sided negative focus of current masculinity

concepts which, in their view, are reductionist and outdated given the changes in masculine images in modern societies. Furthermore, the authors state the need to investigate populations with clinically relevant depression [11].

The current state of research constitutes a weak basis for the development of new therapies for male depression. In particular, we do not know how different patterns of masculinity orientations are distributed in men treated for depression and how these patterns are associated with their clinical status. If a traditional masculinity orientation is a risk factor for depression, we would expect that a majority of male depressed patients hold such normative values. This expectation could be counteracted if men with traditional masculinity orientations were found to stigmatize people with mental disorders and, as a consequence, refuse professional help [17]. The interaction between traditional masculinity orientations, work related attitudes and depression symptom patterns is not well understood. Thus, it is currently unclear how noxious masculinity orientations can be dealt with in depression treatment.

In our study, we aim to address this research gap by identifying different patterns of masculinity and work role orientations in men treated for depression. We further want to investigate the associations of these patterns with stigmatization of mental illness, help-seeking behavior and the severity of psychiatric symptoms,

We expect that the results of our study will contribute to: 1. improving the knowledge about noxious and salutogenetic elements of masculinity and work-role orientations; 2. identifying specific target groups for male depression treatment with reference to types of masculinity and work role orientation; 3. developing adequate treatment techniques or settings for male depressed patients with reference to types of masculinity and work-role orientation.

## Methods

### Study design

This is a cross-sectional observational study on male persons who were treated for depression in psychiatric inpatient or outpatient services, with medication or psychotherapy or both at the time of the study.

### Ethical approval

The study protocol was approved by the ethical board of the Medical Faculty at Ulm University under application number 202/15.

### Sample

Inclusion criteria for study participation were: Male gender, age 18 to 65 years, being in treatment for depression in a psychiatric inpatient or outpatient service.

### Recruitment process

In Germany outpatient treatment for depression is provided by family doctors (general practitioners), psychiatrists and psychological psychotherapists working in private practice while inpatient treatment is provided by general mental hospitals and by psychosomatic hospitals. Hospitals also provide treatment in outpatient clinics or day-hospitals. The treatment in all of these facilities is covered by the statutory health insurance for about 90% of the German population.

Psychiatric and psychosomatic hospitals provide inpatient and outpatient medical and psychotherapeutic treatment for all types of mental illness. Patients with psychotic disorders, addictions, severe depressive episodes and acute suicidal behavior are mainly treated in

psychiatric hospitals. Psychosomatic hospitals are mainly concerned with diseases in which interactions between psychological and physical factors play the central role. These include eating disorders, somatoform disorders and depressive disorders. While psychological psychotherapists are only allowed to provide psychotherapy, office-based family doctors and psychiatrists can provide pharmacological treatment and (in case of adequate qualification) also psychotherapy.

In the attempt to reach male patients with different levels of depression severity and utilizing different types of treatment, we recruited study participants in all facilities mentioned above located in two major cities (Ulm/ Neu-Ulm and Augsburg) with an overall population of about 500.000 inhabitants and in four rural districts (Neu-Ulm, Günzburg, Dillingen and Donauwörth) with a total of about 500.000 inhabitants.

Mental health care professionals from the study regions were informed about the study by a symposium held at Ulm University before study onset. We also distributed information letters to relevant service providers by ordinary mail and e-mail. Furthermore, we initiated a media campaign in the local newspapers to inform about the aims of the study and invite men in current treatment for depression or burnout to participate in the study. Subjects who were interested could contact the study team by phone or e-mail.

Professionals working in outpatient facilities who consented to support the study were provided with information material which they were asked to distribute to eligible patients. The participant information pack included information about the study and an answering form together with a prepaid envelope addressed to the study center. Having returned the study forms, patients were contacted for an appointment by study workers. Patients who agreed to meet the study worker were asked to sign the informed consent form and to complete the study questionnaires. Patients in current inpatient treatment were asked by doctors or other clinical staff whether they were willing to meet the study worker for an information meeting. During the meeting potential study participants were informed about the aims and content of the study, and they were asked to sign the informed consent form and to complete the questionnaires. Male persons who answered to the press campaign were asked if they were currently being treated for depression and 18 to 65 years old. Where this was the case, patients were asked for an appointment with the study worker. During the meeting potential participants were informed about the aims and content of the study and asked to sign the informed consent form and to complete the study questionnaires.

All participants received a sum of 30 € for their participation in the study.

## Assessment instruments and procedures

All assessments were conducted by means of standardized assessment instruments with validated psychometric properties.

## Male Role Norms Scale (MRNS)

The Male Role Norms Scale (MRNS) is a self-rating questionnaire measuring respondents' beliefs about male role norms by asking their agreement or disagreement with 57 statements on a 7-point Likert scale. While the MRNS was originally conceptualized to measure male role orientation along the four dimensions of "achieving status," "cultivating independence and self-confidence," "aggressiveness" and "antifemininity", [18]. Thomson and Pleck [18] identified a three-factor structure including the dimensions "status," "toughness" and "antifemininity." The three-dimensional factorial structure was also confirmed for the German version of the MRNS [19, 20]. Cronbach's alpha as reported for the German Version of the MRNS is .83 for status, .77 for toughness and .81 for antifemininity [19, 20].

### Work-related behavior and experience scale (AVEM)

The Work-Related Behaviors and Experiences Pattern (AVEM = Arbeitsbzogene Verhaltens-
und Erlebensmuster) is a self-rating questionnaire, measuring the following 11 dimensions:
1. subjective priority of work, 2. professional ambition, 3. over-commitment, 4. perfection-
ism, 5. professional distancing ability, 6. resignation tendency, 7. offensive problem manage-
ment, 8. calmness/serenity, 9. experience of professional success, 10. life satisfaction, and 11.
experience of social support. The AVEM consists of 66 items asking for agreement or disagree-
ment with statements on a 5-point Likert scale. Cronbach's alpha values between .79 and .83
are reported for the AVEM subscales [21].

### Depression Stigma Scale (DSS)

The Depression Stigma Scale (DSS) has two subscales measuring negative attitudes about
depression (DSS Personal Stigma) and assumptions about other people's negative attitudes
toward depression (DSS Perceived Stigma) [22, 23]. The original DSS consists of nine state-
ments assessing the respondents' own attitudes and nine statements assessing the respondents'
assumptions about the attitudes of other people by asking for agreement or disagreement on a
5-point Likert scale. For the purpose of our study, we added two items to assess the respon-
dents' gender-related attitudes ("depression is a typical women's disease" and "it is unmanly to
have depression") and two items to assess the respondents' assumptions about other people's
gender-related attitudes about depression ("most people think that depression is a typical
women's disease" and "most people think that it is unmanly to have depression"). Cronbach's
alphas for the original DSS total scale and its subscales were reported as 0.78, 0.76 and 0.82,
respectively [24].

### Patient Health Questionnaire (PHQ-SADS)

The Patient Health Questionnaire, Somatic Anxiety, and Depressive Symptoms (PHQ-SADS)
includes 9 items for assessing depressive symptoms (PHQ-9), 7 items for assessing anxiety
symptoms (GAD-7) and 15 items (PHQ-15) assessing somatic symptoms [24]. Test of the psy-
chometric properties of the PHQ subscales revealed Cronbach's alpha values of 0.89 for the
PHQ-9, 0.92 for the GAD-7 and 0.80 for the PHQ-15 [24]. The PHQ-9 provides cut-off values
for classifying minimal (1–4), mild (5–9), medium (6–14) and severe (15–27) depressive symp-
toms [24]. Since our sample includes only male participants we used the PHQ-15 without one
item asking for menstruation complains.

### Duration of Untreated Illness (DUI)

For estimating the Duration of Untreated Illness (DUI) we asked study participants in which
year they had depressive symptoms for the first time and in which year they had first received
professional treatment for these symptoms. We did not ask for the exact date of symptom or
treatment onset. Therefore, years of treatment delay were calculated by (year of treatment
onset–year of symptom onset).

### Control variables

In order to control for confounding effects we included the following variables because of
their established statistical associations with the measures used for the assessment of masculin-
ity norms and work-role orientations [20, 21]: age, having a partner (no = 0; yes = 1), presence
of children in the household (no = 0; yes = 1), higher education (general university entrance
qualification = Abitur or university degree = 1; other = 0), unemployment (no = 0; yes = 1),

household income above 3.000 € (no = 0; yes = 1), blue-collar work (n = 0; yes = 1); recruitment setting (1 = psychiatric hospital; 2 = psychosomatic hospital; 3 = family doctor; 4 = press invitation). The cut-off value of 3.000 € for household income was used because the mean net household income in Germany in 2016 was about 3.300 € [25].

### Statistical analyses

We conducted a latent profile analysis (LPA) for the identification of latent patterns of associations between male role norms and work-related attitudes and experiences. LPA is the extension of latent class analysis (LCA) for continuous variables. LPA is based on the assumption that statistical associations between the characteristics $x_1 - x_k$ of a sampling unit $i$ (e.g. a person) result from the sampling unit's belonging to the category (class) $k$ of a latent variable $\Theta$. The generalized probability model of the LPA for continuous variables is defined as:

$$f(\boldsymbol{x}_i|\theta) = \sum_{k=1}^{K} \pi_k f_k(x_i|\theta_k)$$

where the term $f(x_i|\theta)$ indicates the class-specific density function of the distribution of the means, variances and covariances of the variables $x_1 - x_k$, and $\pi_k$ indicates the probability of membership of the sampling unit i in the latent class k [26].

Membership probabilities were estimated by means of a maximum likelihood approach with the expectation-maximization (EM) algorithm. Sampling units were assigned to latent classes based on maximal membership probability [26, 27].

Comparative model fit was tested by means of the Akaike information criterion (AIC) and the adjusted Bayes information criterion (ABIC), whereby lower values indicate better fit [28, 29]. The maximum number of classes was determined by means of the likelihood ratio test, indicating whether a model with k classes fits the data significantly better than a model with k–1 classes [27]. The certainty of the class delineation is estimated by the entropy parameter, which ranges from zero to one. Values closer to one indicate better discrimination. LPA was computed using M-Plus 7.2 [30].

For the investigation of sociodemographic and mental health care characteristics related to latent class assignment, we computed a multinomial logistic regression model with assignment to the highest numbered class as reference category, including the control variables listed above.

We conducted analyses of variance (ANOVA) to test mean differences of the items included in the LPA and for symptoms of depression (PHQ-9), anxiety (GAD-7), somatization (PHQ-15), DSS personal stigma, DSS perceived stigma and DUI between latent classes. We tested the equality of variances by means of the Bartlett test and applied Bonferroni's post hoc test in case of equal variances and Scheffe's post hoc test in case of unequal variances for testing mean differences between latent classes.

We computed multiple linear regression models to estimate adjusted means by class membership for symptoms of depression (PHQ-9), anxiety (GAD-7), somatization (PHQ-15), DSS personal stigma, DSS perceived stigma and reported DUI controlling for age, presence of a partner, presence of children, education, unemployment, income, blue-collar work, duration of mental illness and recruitment setting. In addition, all models except that for reported DUI were controlled for reported DUI. We estimated the variance inflation factor (VIF) for each model variable to test for multicollinearity. An average VIF > 10 is regarded as indicating collinearity problems in a linear regression model [31].

We computed a path model to investigate the direct and indirect effects of latent class membership on stigmatization (total DSS score), the reported DUI and the severity of psychiatric

symptoms (total PHQ-SADS score) using the membership to class 3 as reference category. We estimated robust standard errors for path coefficients in order to take account of the skew distribution of the variable indicating DUI. For testing the fit of the path model we estimated the Comparative Fit Index (CFI), the Tucker Lewis Index (TLI), The Root Mean Squared Error of Approximation (RMSEA) and the Standardized Root Mean Square Residual (SMR). A good model fit is indicated by CFI and TLI above 0.95 and RMSEA and SMR below 0.05 [32].

Regression models and path analyses were computed using STATA 15 [33].

## Results

In total, 265 men agreed to participate in the investigation. 15 persons were excluded from the analysis because they were not currently being treated for depression. From the 250 men included in the data analyses (see Table 1), 128 (51.2%) were recruited in psychiatric hospitals, 45 (18.0%) came from psychosomatic hospitals, 6 (2.4%) were asked by family doctors, and 71 (28.4%) study participants had responded to the press invitation. Of the participants who responded to the press invitation 5 (7.7%) were treated by family doctors, 28 (43.1%) were treated by office based psychiatrists, and 25 (38.5%) were treated by office based psychologists. Having ever been treated for depression in a psychiatric or psychosomatic hospital was reported by 31(43.7%) of the participants who responded to the press invitation.

On average, participants were 47 years old (SD = 12.0 years), 37.2% (n = 99) had a higher education, and 38.5% (n = 95) had a monthly household income above the average of the German households (3000 € and higher). About 59.2% (n = 148) lived with a partner and 59.6% (n = 149) lived, with children. While 25.6% (n = 64) of the participants were blue collar workers, 13.2% (n = 33) stated to be unemployed at the time of the interview.

On average, participants had first suffered symptoms of depression 11.5 years (SD = 10.2) earlier but the mean duration of untreated depression was 3.7 years (SD = 6.5 years).

About half of the sample reported symptoms of depression classified as severe based on the PHQ-9 cut-off value. With the exception of the few participants who were recruited from family doctors the distribution of the symptom severity varied only slightly across recruitment settings.

### Results of Latent Profile Analysis (LPA)

Model fit parameters for LPAs with increasing numbers of classes are presented in Table 2. As indicated by the LRT, the three-class model fits the data significantly better than the two classes model, while the model fit does not significantly improve with more than three classes. The AIC, the ABIC and the entropy parameter improve with an increasing number of classes, but the change is greatest between the two- and three-class models.

Based on the comparison of model fit parameters, we selected the three-class model for further consideration.

Fig 1 presents the latent class profiles of the mean distribution of the dimensions of adherence to traditional masculinity and occupational role orientation broken down by class membership in the three-class model (means and SD are presented in S2 Table). The first three categories represent the dimensions of the masculinity role norm scale (MRNS): status, toughness and anti-femininity. Comparisons of means between the three classes indicate that men assigned to class 2 (red line) have significantly higher values on all three dimensions indicating a more traditional male role orientation, while the values of those assigned to class 1 (blue line) and class 3 (green line) do not differ significantly from each other and indicate a less traditional male role orientation.

**Table 1. Sample characteristics by recruitment setting.**

| | Total | Psychiatric hospital | Psychosomatic hospital | Family doctor | Press invitation |
|---|---|---|---|---|---|
| **N (%)** | 250 (100) | 128 (51.2) | 45 (18.0) | 6 (2.4) | 71 (28.4) |
| **Age** mean (SD) | 46.6 (12.0) | 46.7 (12.9) | 43.2 (10.0) | 36.3 (15.2) | 49.4 (10.3) |
| **Higher education** n (%) | 93 (37,2) | 29 (22.7) | 25 (55.6) | 3 (50.0) | 36 (50.7) |
| **Monthly net household income above 3000€** n (%) | 95 (38.0) | 35 (27.3) | 27 (60.0) | 2 (33.3) | 31 (43.7) |
| **Living with partner** n (%) | 148 (59.2) | 72 (56.3) | 31 (68.9) | 2 (33.3) | 43 (60.6) |
| **Living with children** n (%) | 149 (59.6) | 84 (65.6) | 24 (53.3) | 1 (16.7) | 40 (56.3) |
| **Unemployed** n (%) | 33 (13.2) | 19 (14.8) | 5 (11.1) | 0 (0.0) | 9 (12.7) |
| **Blue collar worker** n (%) | 64 (25.6) | 40 (31.3) | 7 (15.6) | 0 (0.0) | 17 (23.9) |
| **Duration of illness in years** mean (SD) | 11.5 (10.2) | 11.8 (10.2) | 8.7 (8.0) | 13.2 (10.5) | 12.8 (11.3) |
| **Duration of untreated illness (DUI) in years** mean (SD) | 3.7 (6.5) | 3.1 (5.9) | 4.7 (7.6) | 3.3 (5.8) | 4.2 (6.8) |
| **Severity of depression (PHQ-9)** | | | | | |
| **Minimal** (1–4) n (%) | 18 (7.2) | 9 (7.0) | 3 (6.7) | 1 (16.7) | 5 (7.4) |
| **Mild** (5–9) n (%) | 47 (18.8) | 23 (18.0) | 8 (17.8) | 0 (0.0) | 16 (22.5) |
| **Medium** (10–14) n (%) | 61 (24.4) | 32 (25.0) | 11 (24.4) | 1 (16.7) | 17 (23.9) |
| **Severe** (15–27) n (%) | 124 (49.6) | 64 (51.1) | 23 (51.1) | 4 (66.7) | 33 (46.5) |

The comparison of the dimensions representing job-related attitudes reveals that men assigned to class 2 reported the highest levels of job priority, occupational ambition, over-commitment and perfectionism, followed by those men assigned to class 3. Men assigned to class 1 reported the lowest level of these attitudes. Compared to men assigned to classes 1 and 3, men assigned to class 2 reported the lowest ability to distance themselves from their jobs while also showing the greatest tendency to give up when confronted with professional problems. In contrast, men assigned to class 3 reported the lowest likelihood of resignation and the highest level of "coping with professional problems in an offensive way". Men assigned to class 3 reported the highest level of serenity in their professional context, and they also reported the highest level of "experiencing success, social support and life satisfaction" compared to those assigned to classes 1 and 2. While men assigned to class 2 had the lowest level of serenity, men assigned to class 1 reported the lowest level of experiencing job success. Men assigned to classes 1 and 2 did not differ with regard to life satisfaction or the experience of social support.

In summary LPA results indicate that:

- Men assigned to class 1 reported a less pronounced adherence to traditional masculinity norms combined with low occupational ambitions and a low ability to manage professional stress and to experience professional success and recognition.

- Men assigned to class 2 reported a strong adherence to traditional masculinity norms combined with high professional ambitions but a low ability to manage professional stress and to experience professional success and recognition.

- Men assigned to class 3 reported a less pronounced adherence to traditional masculinity norms in combination with medium professional ambitions and a high ability to manage professional stress and to experience professional success and recognition.

## Associations of latent class assignment with sociodemographic and treatment characteristics

The between class differences of sociodemographic characteristics and recruitment setting are provided in Table 3.

**Table 2. Model fit parameter for the LPA model selection.**

| Model | classes k | Log-likelihood | LRT p k-1 | AIC | ABIC | Entropy |
|---|---|---|---|---|---|---|
| 1 | 2 | -10702.648 | 0.2720 | 21079.494 | 21094.603 | 0.821 |
| **2** | **3** | **-10496.747** | **0.0231** | **20809.894** | **20830.274** | **0.843** |
| 3 | 4 | -10346.947 | 0.3802 | 20702.015 | 20727.665 | 0.847 |
| 4 | 5 | -10233.093 | 0.3621 | 20642.187 | 20673.108 | 0.859 |

LRT = Log-likelihood ratio test; AIC = Akaike information criterion; ABIC = Adjusted Bayes Information Criteria

As revealed by the risk ratios (RR) resulting from the multinomial logistic regression model (Table 3) the probability of assignment to latent class 1 in comparison to assignment to class 3 was lower for participants who had a higher education, and for those who lived with a partner, but higher for participants who were unemployed. In comparison to participants who were recruited in psychiatric hospitals those who were recruited in psychosomatic hospitals or those who were recruited by press invitation had a higher probability to become assigned to class 1 rather than class 3.

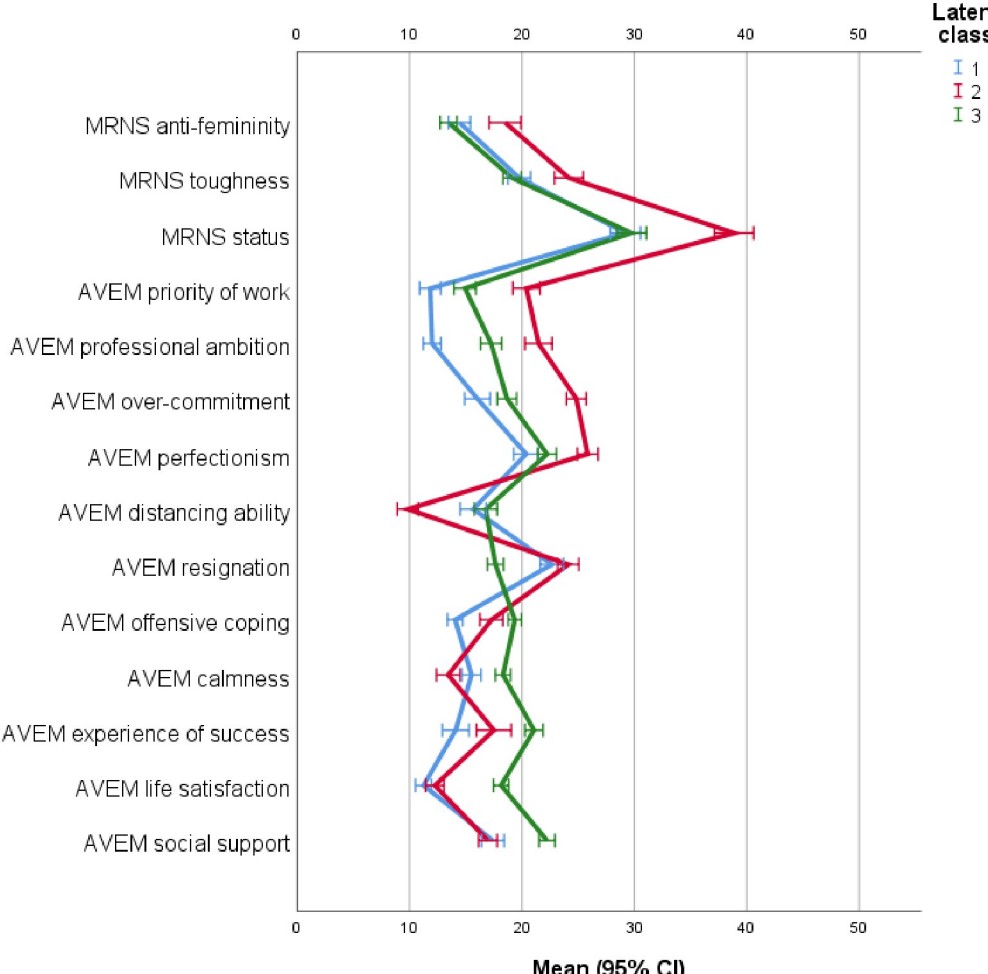

**Fig 1. Latent class profile for masculinity norms and work role orientations.** Means and 95% confidence intervals.

**Table 3. Multi-nomial logit regression model of latent class assignment on sociodemographic characteristics, duration of illness and recruitment setting.**

| N (%) | Class 1 85 (34.0) | Class 2 58 (23.2) | Class 3 107 (42.8) | RR[1] Class 1 vs Class 3 (p) | RR[1] Class 2 vs Class 3 (p) |
|---|---|---|---|---|---|
| **Age** mean (SD) | 48.5 (11.4) | 45.5 (11.9) | 45.7 (12.4) | 1.03 (0.148) | 0.99 (0.524) |
| **Higher education** n (%) | 22 (25.9) | 16 (27.6) | 55 (51.4) | **0.28 (0.003)** | **0.27 (0.006)** |
| **Monthly net household income above 3000€** n (%) | 20 (23.5) | 22 (37.9) | 53 (49.5) | 0.56 (0.170) | 0.98 (0.964) |
| **Living with partner** n (%) | 36 (42.4) | 33 (56.9) | 79 (73.8) | **0.21 (0.009** | **0.35 (0.012)** |
| **Living with children** n (%) | 48 (56.5) | 41 (70.7) | 60 (56.1) | 1.58 (0.296) | **3.67 (0.005)** |
| **Unemployed** n (%) | 18 (21.2) | 6 (10.3) | 9 (8.4) | **2.95 (0.045)** | 1.10 (0.871) |
| **Blue collar worker** n (%) | 28 (32.9) | 18 (31.0) | 18 (16.8) | 1.35 (0.492) | 1.98 (0.136) |
| **Duration of illness in years** mean (SD) | 13.5 (11.3) | 12.1 (11.3) | 9.6 (8.3) | 1.05 (0.075) | 1.01 (0.737) |
| **Duration of untreated illness (DUI) in years** mean (SD) | 3.3 (5.8) | 5.7 (8.7) | 3.0 (5.3) | 0.94 (0.150) | 1.05 (0.219) |
| **Recruitment setting** | | | | | |
| Psychiatric hospital n (%) | 40 (47.6) | 27 (46.6) | 61 (57.0) | Ref. | Ref. |
| Psychosomatic hospital n (%) | 14 (16.5) | 13 (22.4) | 18 (16.8) | **4.43 (0.005)** | **3.68 (0.013)** |
| Family doctor n (%) | 2 (2.4) | 1 (1.7) | 3 (2.8) | 2.09 (0.470) | 1.32 (0.831) |
| Press invitation n (%) | 29 (34.1) | 17 (29.3) | 25 (23.4) | **3.64 (0.004)** | **3.06 (0.018)** |

[1] Multi-nomial logit regression model with assignment to Class 3 as reference category.

The probability of assignment to class 2 vs. class 3 was lower for participants with a higher income and for those who lived with a partner, but higher for those who had children. In comparison to participants who were recruited in psychiatric hospitals those who were recruited from psychosomatic hospitals or those who were recruited by press invitation had a higher probability to become assigned to class 2 rather than class 3. A pseudo-$R^2$ of 0.16 indicates a sufficient fit of the multinomial logit model.

### Symptoms of mental disorder, stigmatization and duration of untreated illness by latent class membership

The adjusted means (see S3 Table for raw means) for the PHQ subscales in Fig 2 indicate that participants assigned to class 3 had significantly fewer symptoms of anxiety and depression than those assigned to classes 1 and 2 while the means of somatization symptoms differed only between classes 2 and 3. Results of the VIF estimation reveal an average VIF of 1.47 for all regression models which is far below the critical value of 10. The highest VIF was estimated for reported duration of depressive illness with 2.18 which is also far below the critical value of 10.

### Path analyses

Fig 3 reveals the standardized path coefficients (ß) and robust standard errors (se) of latent class membership in classes 1 and 2 vs. membership in class 3 on stigmatizing attitudes, reported DUI and symptoms of mental disorder.

As indicated by path coefficients, study participants assigned to class 1 reported significantly more stigmatizing attitudes (ß = 0.28; se = 0.06; p < 0.001) and significantly more severe symptoms of mental disorder (ß = 0.33; se = 0.056; p < 0.001) compared to those assigned to class 3. The path coefficient between class 1 membership and DUI is not significant (ß = -0.022; se = 0.063; p = 0.729).

Study participants assigned to class 2 reported significantly more stigmatizing attitudes (ß = 0.43; se = 0.06; p < 0.001) and more severe symptoms of mental disorder (ß = 0.4; se = 0.067;

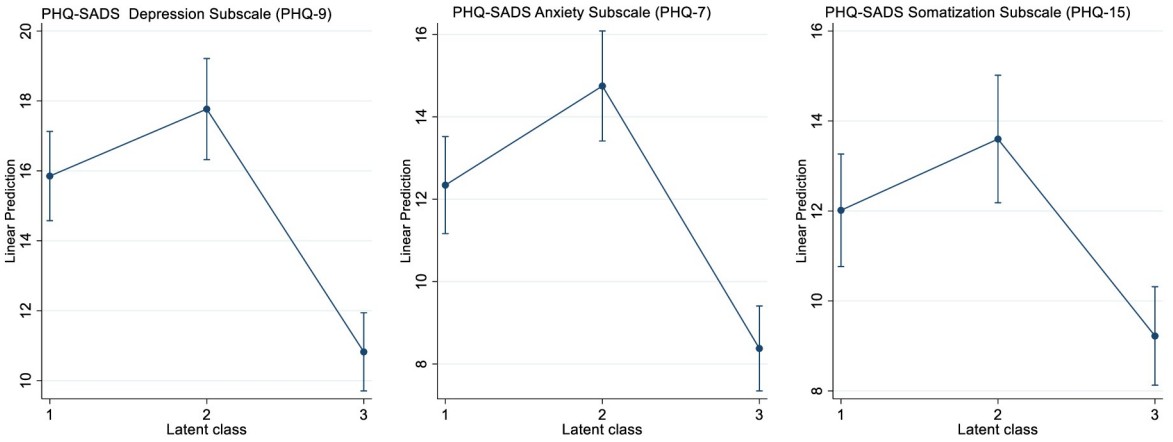

**Fig 2. Patient Health Questionnaire (PHQ-SADS) subscales by latent class assignment.** Means and 95% confidence intervals adjusted for age, partner, children, higher education, unemployment, bluecollar worker, income, duration of illness, duration of untreated illness, recruitment setting.

p = 0.001) than those assigned to class 3, while the path from class 2 to DUI (ß = 0.13; se = 0.072; p = 0.091) indicates no significant effect.

While stigmatizing attitudes were significantly related to an increased DUI (ß = 0.1; se = 0.016; p < 0.001) and to increased severity of symptoms (ß = 0.17; se = 0.06; p < 0.01) the path coefficient from DUI to symptoms of mental disorder (ß = -0.07; se = 0.06; p < 0.261) indicate no significant effect.

As indicated by the decomposition of direct and indirect effects, membership to class 1 (ß = 0.02; p < 0.001) and membership to class 2 (ß = 0.044; p = 0.001) were both indirectly related to a longer DUI compared to membership to class 3.

The model explains 3.8% of the variance of DUI but 16% of the variance of stigmatizing attitudes and 24% of the variance of symptoms of mental disorder. The model fit parameters (CFI = 1.0; TLI = 1.0; RMSEA = 0.000; SRMR = 0.006) indicate a very close fit of the model with the empirical covariance structure.

## Discussion

Our findings confirm previous results that the adherence to traditional masculinity concepts is directly associated with more stigmatizing attitudes against depressive illness and increased symptoms of mental disorder [10, 11, 34, 35]. However, our results also indicate that men treated for depression are not a homogenous group with respect to their adherence to traditional masculinity norms and occupational role orientations. Furthermore, our findings reveal that a less pronounced adherence to traditional masculinity is only associated with better mental health and less stigmatizing attitudes if it coincides with a medium level of occupational ambition and adequate abilities to cope with occupational stress and to perceive recognition at work.

These results underline the interconnectedness between masculinity concepts and occupational role orientations for male mental health [36]. In western societies professional success and social status are core elements of the traditional male gender role [37]. The workplace therefore can become both an important generator of feelings of personal accomplishment and social recognition but also a source of emotional strain and the risk of failure [38–40]. Our

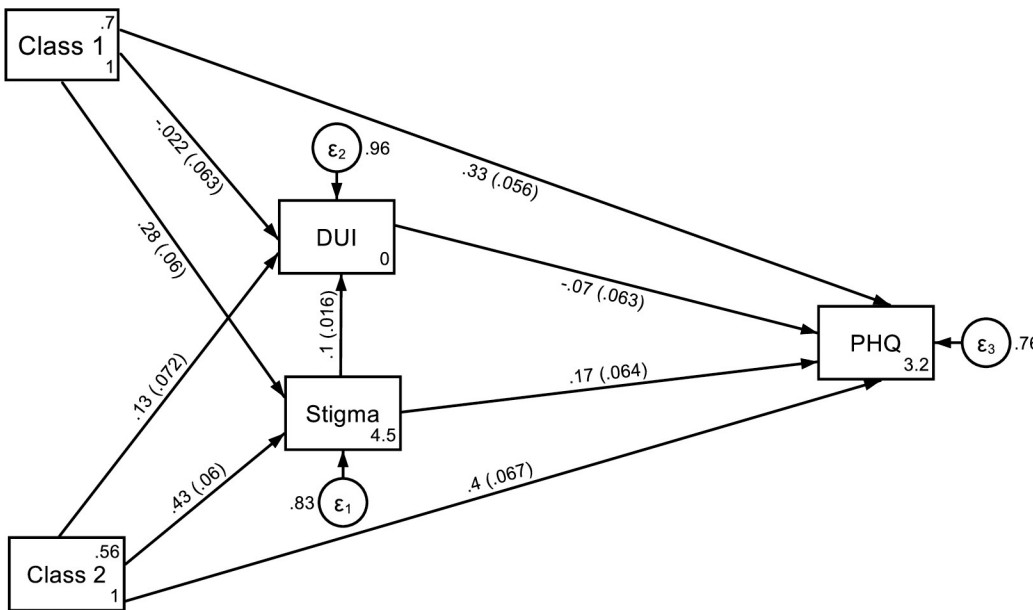

**Fig 3. Standardized path coefficients and robust standard errors of the effects of assignment to latent Class1 or latent Class 2 vs. latent Class 3 (reference category) on stigmatizing attitudes (stigma = DSS total sum score), the duration of untreated illness (DUI), and symptoms of mental disorder (PHQ = PHQ-SADS total sum score).**

data suggest that those study participants who adhere most strongly to the traditional masculine role orientations have the highest levels of professional ambition and the lowest capacity to cope with occupational stress. Research on occupational mental health revealed that job over-commitment is an intrinsic risk factor increasing the negative effects of job-strain on mental health [39].

Under such circumstances keeping a distance from occupational ambitions could ease emotional strain and fear of failure [39]. Further research should investigate, whether this relief comes at the price of losing crucial emotional rewards, e.g. experiencing accomplishment and success. Both pronounced work role adherence and radical distance appear to be related to poorer emotional well-being and more stigmatizing attitudes (when compared with medium levels of occupational ambition and effort).

With regard to reported DUI, we found only indirect effects of traditional masculinity and work-role orientations indicating that these effects were mediated by stigmatizing attitudes. This is in accordance with the finding of a recent review article of 27 studies [41] that negative attitudes toward people with mental illness were associated with lower willingness to seek professional help. On the basis of an elaborated theoretical model Schomerus et al. [17] found that with increasing level of stigmatizing attitudes people with untreated mental illness were more reluctant to identify themselves as being mentally ill and to accept the need for mental health care.

Our finding, of no association between reported DUI and symptom severity is in accordance with previous studies revealing that longer DUI is related to reduced treatment effectiveness and a lower likelihood of remission but not to symptom severity and other clinical features [42–46].

## Limitations

Our study sample includes only men currently treated for depression. Therefore, we cannot draw any conclusions about the representativity of our study results. Therefore, the finding

that only 23% of study participants had a traditional masculinity orientation may reflect that men with a lower adherence to traditional masculinity norms are more prepared to seek professional help.

The recruitment of participants via press invitation could result in a further limitation of generlizability due to self-selection bias associated with peoples' previous experiences with treatment or other clinical or personal characteristics.

In addition, our measurement of DUI is based on self-report only and therefore susceptible to recall bias.

Finally, due to the cross-sectional design our study does not allow to make any conclusions about causal effects or processes of change.

## Conclusions

Overall, this study reveals that masculinity norms and work-role orientations are relevant among men treated for depression. Our results suggest that any slackening of traditional masculinity norms may be related to better psychological well-being when it not coincides with a complete distancing from work-role ambitions and/or deficits in coping with occupational stress. Longitudinal studies are needed to clarify whether helping depressed men (a) to take a distance from traditional masculinity norms, (b) to enhance a balanced work-role orientation, (c) to develop effective stress coping, and (d) to reduce stigmatizing attitudes could improve the effectiveness of depression treatment.

## Supporting information

**S1 Fig. Depression Stigma Scale (DSS) subscales and duration of untreated illness (DUI) by latent class assignment.** Means and 95% confidence intervals adjusted for age, partner, education, unemployment, income, duration of illness, duration of untreated illness (only in the DSS models), recruitment setting.
(DOCX)

**S1 Table. Cronbach's alpha of the study instruments in the current sample.**
(DOCX)

**S2 Table. Means and standard deviations of the dimensions of the Male Role Norms Scale (MRNS) and the work related behavior and experience scale (AVEM) by latent class assignment.**
(DOCX)

**S3 Table. Raw mean differences of PHQ-SADS subscales, DSS subscales and DUI by latent class assignment.**
(DOCX)

**S1 File.**
(TXT)

**S1 Data.**
(DAT)

**S2 Data.**
(STS)

## Author Contributions

**Conceptualization:** Reinhold Kilian, Petra Beschoner, Harald Gündel, Thomas Becker, Silvia Krumm.

**Data curation:** Annabel Müller-Stierlin, Maja Stiawa, Maria Panzirsch.

**Formal analysis:** Reinhold Kilian, Annabel Müller-Stierlin, Maria Panzirsch, Silvia Krumm.

**Funding acquisition:** Reinhold Kilian, Harald Gündel, Thomas Becker, Silvia Krumm.

**Investigation:** Felicitas Söhner, Petra Beschoner, Tobias Staiger, Maja Stiawa, Karel Frasch, Maria Panzirsch, Silvia Krumm.

**Methodology:** Reinhold Kilian, Petra Beschoner, Harald Gündel, Maria Panzirsch, Silvia Krumm.

**Project administration:** Annabel Müller-Stierlin, Petra Beschoner, Harald Gündel, Silvia Krumm.

**Supervision:** Harald Gündel, Thomas Becker, Karel Frasch, Max Schmauß, Silvia Krumm.

**Visualization:** Reinhold Kilian.

**Writing – original draft:** Reinhold Kilian, Annabel Müller-Stierlin, Silvia Krumm.

**Writing – review & editing:** Reinhold Kilian, Annabel Müller-Stierlin, Felicitas Söhner, Petra Beschoner, Harald Gündel, Tobias Staiger, Maja Stiawa, Thomas Becker, Karel Frasch, Maria Panzirsch, Max Schmauß, Silvia Krumm.

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
