## [Decision Letter · Decision Letter 0]

13 Dec 2019

PONE-D-19-27414

Why do men hesitate to seek professional treatment for depression? The influences of masculinity norms and Stigmatizing attitudes on the delay of the search for professional help by men with depression

PLOS ONE

Dear Dr. Kilian,

Thank you for submitting your manuscript to PLOS ONE. After careful consideration, we feel that it has merit but does not fully meet PLOS ONE’s publication criteria as it currently stands. Therefore, we invite you to submit a revised version of the manuscript that addresses the points raised during the review process.

We would appreciate receiving your revised manuscript by January 7, 2020. To enhance the reproducibility of your results, we recommend that if applicable you deposit your laboratory protocols in protocols.io, where a protocol can be assigned its own identifier (DOI) such that it can be cited independently in the future. For instructions see: http://journals.plos.org/plosone/s/submission-guidelines#loc-laboratory-protocols

We look forward to receiving your revised manuscript.

Kind regards,

Stephan Doering, M.D.

Academic Editor

PLOS ONE

Journal Requirements:

Reviewers' comments:

Reviewer's Responses to Questions

**Comments to the Author**

1. Is the manuscript technically sound, and do the data support the conclusions?

Reviewer #1: Partly

Reviewer #2: Partly

2. Has the statistical analysis been performed appropriately and rigorously? 

Reviewer #1: I Don't Know

Reviewer #2: I Don't Know

3. Have the authors made all data underlying the findings in their manuscript fully available?

Reviewer #1: No

Reviewer #2: No

4. Is the manuscript presented in an intelligible fashion and written in standard English?

Reviewer #1: Yes

Reviewer #2: Yes

5. Review Comments to the Author

Reviewer #1: Introduction

The question of the role of masculinity in delaying treatment seeking is an important one, however the study as presented does not focus sufficiently on this. In both the introduction and the discussion the focus is much more on male role orientations and work roles in relation to depression. The problem with that focus is that the sample are all in treatment for depression and there is no non-depressed comparison group. Thus, all the study can really examine is different configurations of masculinity that can be discerned among depressed men. The authors need to make a much stronger case about why this warrants attention, and re-title the paper to reflect that focus. I believe the question of masculinity and delay in help-seeking, and the role of stigma in that, is of much more interest and if the paper could be refocused around that it would be of more value.

The inclusion of ‘burnout’ is at no point explained – is it a proxy for poor mental health? Or a condition in its own right? Or a recruiting convenience? It needs to be discussed in the introduction and a rational provided for combining a burnout sample with a depressed sample.

Methods.

The paper doesn’t report on the qualitative element of the study so I don’t think it necessary to present the paper as based on a mixed-methods design.

Can the cities or towns where the study took place be named in addition to the country and the years of the study.

The inclusion of ‘burnout’ as an inclusion criteria needs much more explanation – i.e. what is ‘burnout’, how is it diagnosed, what is the relationship between burnout and depression – basically we need sufficient information to be able to judge that combining these two samples provides a suitable sample for the analysis undertaken and the conclusions drawn.

We need more information on participants recruited through newspaper advertisements – how were they screened.

We need more information on the format of data collection – was it self-complete questionnaires, was it by face-to-face interview? Was it conducted by clinic staff or by the researchers etc.

Delay in psychiatric treatment – notes that delay is calculated based on “the year in which a psychiatric treatment for depressive symptoms was used for the first time”. In table 1 delay is reported as ‘months untreated’ – need to explain how those months were calculated i.e. from January in the year treatment was initiated?

Also, if receiving treatment for burnout is an eligibility criterion is it possible that some of the men might never have been treated for depression?

Overall, we need more clarity around the diagnostic status of the sample at recruitment. This also needs to be reported in Table 1.

Results.

It would be good to see descriptive statistics for the various measures in Table 1 as well. And as mentioned above, diagnostic status (at minimum if they are being treated for depression or for burnout’.

The text describing figure 5 appears to have errors – the second sentence compares class 2 to class 2. There are also some spelling errors in that paragraph.

Discussion

The discussion would benefit from being more focused on the question that the paper describes in the title and abstract – the relationship between masculine norms and work role and the delaying of treatment seeking, and the putative role of stigma in that relationship. The focus is predominantly on differences in adherence to traditional masculinity and occupation role orientations in a depressed/burnout sample, and the authors have not made a strong argument about why this is important to study.

Much of the discussion then describes the relationship of masculinity and work role orientation which is not relevant to the original research question. Moreover the assumption made “that traditional masculine role orientation is not harmful per se to mental health, but depends on its functional and contextual meaning in the work environment” cannot be supported by the study as the sample is comprised of only men with depression.

Reviewer #2: Overall, the paper paper is focused on an important gap in the literature and makes a contribution to the field. However, there were some limitations which must be seriously taken into account. These include generalizability of the findings without more information about recruitment process and sample, types of occupations of the participants, as well as the recall bias that may be affecting one of the main outcomes (delay in treatment). The study can only provide information on men who ultimately seek care, and there may be a much wider pool of men who never seek treatment at all, who may not be represented in these results and this should be acknowledged.

There is some convoluted writing and significant typos. It may be useful to have the paper copy edited (perhaps professionally) for standard English when revising the paper.

Abstract:

• More detail is needed on the methods. How were patients recruited? Convenience sample? What measures were used? Are they validated measures? Was the study quantitative or qualitative? What was the participation rate? Some of this information can be provided solely in the manuscript, but more detail is needed in the abstract as well.

• Last sentence of the conclusions should read: “in the development of strategies to reduce the delay…and to tailor the psychiatric treatment…”

Introduction:

• The introduction needs to be more focused on the importance of the work environment earlier. As the introduction currently reads, it is not clear immediately that the focus will be occupational ‘identity’ or norms.

• Page 3, last sentence of the first paragraph: “negatively associated with depressive symptoms” is unclear. Can say, “was associated with lower depressive symptoms,” or other language to make the association clearer.

• Page 4, first sentence: Please specify what the ‘control populations’ were. Individuals without depression?

Methods:

• The fact that this is a mixed-methods study needs to be in the abstract.

• Please specify whether the measures were validated measures from the very beginning (first paragraph of study design section).

• There are multiple typos in the methods section.

• When you say that a ‘sub-sample of study participants were selected on the bases if the results’ please specify what the basis of this decision was. Will help the reader understand generalizability of the results. Who was chosen to be interviewed?

• Why was this age rage chosen? Please justify the rationale. Was the idea to choose only working-age men? If so, please specify.

• For some readers, such as myself, we are unfamiliar with the term ‘psychosomatic hospitals.’ Please provide a bit more background on these.

• The paper should specify how many patients came from each of the different settings, as again, this will affect generalizability.

• Please provide more info on the recruitment process. Were patients asked to participate if they were having depressive symptoms, if they were in treatment for depression, etc. What did the recruitment messaging say?

• It says that eligible patients were contacted by doctors or clinical staff about willingness to participate. Was this the case even for participants who came in via the press release?

• Cronbach’s alpha numbers are part of results, not methods. Please move these to the results section.

• Please provide the full name for all acronyms (such as AVEM on page 6).

• Some questions were added to validated scales. This would be unvalidated measures unless validated elsewhere. Please specify if that was the case.

• For ‘delay of treatment,’ which is an important outcome in this study, please provide further information on how this was assessed. There seems to be a lot of room for recall bias, which is inherently problematic and should be acknowledged in the limitations section.

• The information on the LPA could potentially be included in an appendix if allowed, rather than in the body of the paper.

• ‘one-way analyses of variance’= ANOVA and should be specified.

Results:

• Many of the covariates were not described in the methods. How is ‘higher education’ defined? College degree? More than college degree? Duration of illness is lacking units. Is this years, months, etc. Why was the cut-off of >3000 euros used? Please provide the rationale for this decision.

• Page 13, last sentence: the sentence structure is very convoluted, which is obscuring the meaning of the sentence. Do you mean, ‘participants in latent class 1 were less likely to have a partner…than participants in latent class 3’?

• Page 15, last paragraph, second sentence: class 2 is compared to class 2.

• The authors seem to be presenting an overwhelming amount of results that are hard to interpret. I would consider trying to figure out what the main results and take-away findings are and moving all others to an appendix section.

• Many typos throughout the results section.

Discussion:

• The first sentence of the discussion section was not well set up in the introduction. It was unclear that this was the hypothesis that the authors were challenging.

• The bullets in the discussion section are more re-hashing of the results and are actually much clearer summaries than the ones provided in the actual results section.

• ‘Sex’ and ‘gender’ are used interchangeably throughout the paper, when the authors are in fact referring to ‘gender’ only throughout.

• Page 19: there should be something in the discussion about how the info from the study could be used in practice.

Limitations section:

• Self-selection of participants is an important limitation, as it limits the generalizability of results. Can only speak for patients IN treatment who were willing to participate in the study. Cannot speak for men who do not seek treatment at all or who are unwilling to participate in the study. Generalizability should be listed as a limitation specifically.

• It is also a limitation that delay is all self-report, which is highly subjected to recall bias. Unclear how severity of illness, duration of treatment, etc may impact recall of treatment delay.

• A major limitation is that type of occupation is not included in the study results. What type of work did the men included in the study do? Blue collar vs white collar? This seems like it could strongly affect or influence the findings.

Conclusions:

• Can unfortunately not speak about the ‘treatment process,’ only ‘seeking treatment’ or ‘access’ to treatment.

• The last sentence, while true, seems unrelated to the study. More could be said specifically about the impact or use of the findings.

6. PLOS authors have the option to publish the peer review history of their article (what does this mean?). If published, this will include your full peer review and any attached files.

Reviewer #1: No

Reviewer #2: No

---

## [Author Response · Author response to Decision Letter 0]

20 Feb 2020

Reviewers' comments:

Reviewer's Responses to Questions

Comments to the Author

1. Is the manuscript technically sound, and do the data support the conclusions?

Reviewer #1: Partly

Reviewer #2: Partly

2. Has the statistical analysis been performed appropriately and rigorously? 

Reviewer #1: I Don't Know

Reviewer #2: I Don't Know

3. Have the authors made all data underlying the findings in their manuscript fully available?

Reviewer #1: No

Reviewer #2: No

4. Is the manuscript presented in an intelligible fashion and written in standard English?

Reviewer #1: Yes

Reviewer #2: Yes

5. Review Comments to the Author

Reviewer #1: Introduction

The question of the role of masculinity in delaying treatment seeking is an important one, however the study as presented does not focus sufficiently on this. In both the introduction and the discussion the focus is much more on male role orientations and work roles in relation to depression. The problem with that focus is that the sample are all in treatment for depression and there is no non-depressed comparison group. Thus, all the study can really examine is different configurations of masculinity that can be discerned among depressed men. The authors need to make a much stronger case about why this warrants attention, and re-title the paper to reflect that focus. I believe the question of masculinity and delay in help-seeking, and the role of stigma in that, is of much more interest and if the paper could be refocused around that it would be of more value.

Authors' reply

We agree with the reviewer that the main focus of the manuscript is the association between masculinity and work-role orientation with psychiatric symptoms, stigmatization and the delay of treatment in men already treated for depression. In our view the investigation of this question is important because it can improve our understanding of how relevant aspects of masculinity are in the treatment process. 

We changed the title of the manuscript to make our intention more clear and we tried to focus the introduction.

Reviewer

The inclusion of ‘burnout’ is at no point explained – is it a proxy for poor mental health? Or a condition in its own right? Or a recruiting convenience? It needs to be discussed in the introduction and a rational provided for combining a burnout sample with a depressed sample.

Authors' reply

The main reason for including people with burnout was that we suspected that particularly by family doctors the diagnosis of burnout will often be used when they guess that patients refuse to accept the diagnosis of depression. We therefore expected that by including patients treated for burnout we would get access to patients who are reluctant to accept a diagnosis of a mental disorder. However, due to the fact that only very few patients (n = 15) treated for burnout participated in the study, we decided to exclude these patients from our analyses to get a more homogenous sample. We re-calculated all statistical analyses on the basis of the new sample. 

Reviewer

Methods.

The paper doesn’t report on the qualitative element of the study so I don’t think it necessary to present the paper as based on a mixed-methods design.

Authors' reply

The study was conceptualized as a mixed-method study in the sense that the results of the standardized quantitative part of the investigation should be used as the basis for the selection of study participants for further qualitative inquiries. However, we agree with the reviewer that the current manuscript reports only results of the standardized quantitative part of the study. Therefore, we followed the recommendation not denoting the study as mixed-method in this manuscript.

Reviewer

Can the cities or towns where the study took place be named in addition to the country and the years of the study.

Authors' reply

According to the recommendation of reviewer 1 we named the main cities and described the study region more detailed in the method section.

Reviewer

The inclusion of ‘burnout’ as an inclusion criteria needs much more explanation – i.e. what is ‘burnout’, how is it diagnosed, what is the relationship between burnout and depression – basically we need sufficient information to be able to judge that combining these two samples provides a suitable sample for the analysis undertaken and the conclusions drawn.

Authors' reply

See above.

Reviewer

We need more information on participants recruited through newspaper advertisements – how were they screened.

Authors' reply

We described the recruitment process more detailed and we present an extra table with the description of sample characteristics broken down by recruitment setting.

Reviewer

We need more information on the format of data collection – was it self-complete questionnaires, was it by face-to-face interview? Was it conducted by clinic staff or by the researchers etc.

Authors' reply

We described the data collection more detailed in the method section.

Reviewer

Delay in psychiatric treatment – notes that delay is calculated based on “the year in which a psychiatric treatment for depressive symptoms was used for the first time”. In table 1 delay is reported as ‘months untreated’ – need to explain how those months were calculated i.e. from January in the year treatment was initiated?

Authors' reply

We asked patient in which year they had depressive symptoms for the first time and in which year they had their first professional treatment contact and computed the delay of treatment by calculating the difference between both years. We agree with the evaluator that the use of months as a unit of treatment delay suggests a higher measurement accuracy than was actually implemented. We therefore describe the assessment of the variable more detailed and changed the unit of measurement into years.

Reviewer

Also, if receiving treatment for burnout is an eligibility criterion is it possible that some of the men might never have been treated for depression?

Authors' reply

See above.

Reviewer

Overall, we need more clarity around the diagnostic status of the sample at recruitment. This also needs to be reported in Table 1.

Authors' reply

We reported the clinical status of the study participants by means of the PHQ-9 cut of values for the severity of depression in table 1.

Reviewer

Results.

It would be good to see descriptive statistics for the various measures in Table 1 as well. And as mentioned above, diagnostic status (at minimum if they are being treated for depression or for burnout’.

Authors' reply

In addition to the PHQ-9 cut-off values we reported the means and standard deviations of all PHQ subscales broken down by latent-class in table 2.

reviewer

The text describing figure 5 appears to have errors – the second sentence compares class 2 to class 2. There are also some spelling errors in that paragraph.

Authors' reply

We are sorry for this error and corrected the formulation.

Reviewer

Discussion

The discussion would benefit from being more focused on the question that the paper describes in the title and abstract – the relationship between masculine norms and work role and the delaying of treatment seeking, and the putative role of stigma in that relationship. The focus is predominantly on differences in adherence to traditional masculinity and occupation role orientations in a depressed/burnout sample, and the authors have not made a strong argument about why this is important to study.

Authors' reply

We agree that our study is mainly focused on the relationship of masculinity and work-role orientations with the psychological well-being of depressed patients. We therefore tried to underline the clinical relevance of these associations for depression therapy in the introduction and in the discussion. 

Reviewer

Much of the discussion then describes the relationship of masculinity and work role orientation which is not relevant to the original research question. Moreover the assumption made “that traditional masculine role orientation is not harmful per se to mental health, but depends on its functional and contextual meaning in the work environment” cannot be supported by the study as the sample is comprised of only men with depression.

Authors' reply

We agree that our study design and the study sample does not allow general conclusions about the effects of masculinity norms and work-role orientations on the etiology of depressive symptoms and we therefore focused our conclusion to the population of men with depression which is represented by our sample

Reviewer 2

Reviewer #2: Overall, the paper paper is focused on an important gap in the literature and makes a contribution to the field. However, there were some limitations which must be seriously taken into account. These include generalizability of the findings without more information about recruitment process and sample, types of occupations of the participants, as well as the recall bias that may be affecting one of the main outcomes (delay in treatment). The study can only provide information on men who ultimately seek care, and there may be a much wider pool of men who never seek treatment at all, who may not be represented in these results and this should be acknowledged.

Authors' reply

We addressed the limitation of generalizability in the limitation section.

Reviewer

There is some convoluted writing and significant typos. It may be useful to have the paper copy edited (perhaps professionally) for standard English when revising the paper.

Authors' reply

We have completely revised the language of the manuscript, including professional editing

Reviewer

Abstract:

• More detail is needed on the methods. How were patients recruited? Convenience sample? What measures were used? Are they validated measures? Was the study quantitative or qualitative? What was the participation rate? Some of this information can be provided solely in the manuscript, but more detail is needed in the abstract as well.

Authors' reply

We described the study methods in the abstract more detailed.

Reviewer

• Last sentence of the conclusions should read: “in the development of strategies to reduce the delay…and to tailor the psychiatric treatment…”

Authors' reply

We thank the reviewer for this advice and made the recommended changes.

Reviewer

Introduction:

• The introduction needs to be more focused on the importance of the work environment earlier. As the introduction currently reads, it is not clear immediately that the focus will be occupational ‘identity’ or norms.

Authors' reply

We reformulated the introduction to make the focus on masculinity and work-role orientations more clear.

reviewer

• Page 3, last sentence of the first paragraph: “negatively associated with depressive symptoms” is unclear. Can say, “was associated with lower depressive symptoms,” or other language to make the association clearer.

Authors' reply

We thank the reviewer for this advice and made the recommended changes.

Reviewer

• Page 4, first sentence: Please specify what the ‘control populations’ were. Individuals without depression?

Authors' reply

We added the missing information about the control population.

Reviewer

Methods:

• The fact that this is a mixed-methods study needs to be in the abstract.

Authors' reply

Since the manuscript provides only results from the quantitative part of the study, we followed the recommendation of reviewer 1 and deleted the note on the mixed-method character of the study from the manuscript.

Reviewer

• Please specify whether the measures were validated measures from the very beginning (first paragraph of study design section).

Authors' reply

Done.

Reviewer

• There are multiple typos in the methods section.

Authors' reply

We regret the negligence and have tried to correct all mistakes.

Reviewer

• When you say that a ‘sub-sample of study participants were selected on the bases if the results’ please specify what the basis of this decision was. Will help the reader understand generalizability of the results. Who was chosen to be interviewed?

Authors' reply

For only quantitative data were used in this manuscript and the qualitative part of the study had no influence on the quantitative results, we followed the recommendation of reviewer 1 and deleted all notes on the qualitative part of the study from this manuscript.

Reviewer

• Why was this age rage chosen? Please justify the rationale. Was the idea to choose only working-age men? If so, please specify.

Authors' reply

Yes, because our particular interest in work-role orientations we included only men below the usual age of retirement, which in Germany is 65 years.

Reviewer

• For some readers, such as myself, we are unfamiliar with the term ‘psychosomatic hospitals.’ Please provide a bit more background on these.

Authors' reply

We specified the difference between psychiatric and psychosomatic hospitals in the method section.

Reviewer

• The paper should specify how many patients came from each of the different settings, as again, this will affect generalizability.

Authors' reply

We presented detailed information about the recruitment in table 1.

Reviewer

• Please provide more info on the recruitment process. Were patients asked to participate if they were having depressive symptoms, if they were in treatment for depression, etc. What did the recruitment messaging say?

Authors' reply

We presented a more detailed description of the recruitment process and the press invitation in the method section.

Reviewer

• It says that eligible patients were contacted by doctors or clinical staff about willingness to participate. Was this the case even for participants who came in via the press release?

Authors' reply

We added the information that patients who reported on the press release were directly contacted by the study workers.

Reviewer

• Cronbach’s alpha numbers are part of results, not methods. Please move these to the results section.

Authors' reply

Cronbach’s Alpha computed with our data are now presented in the result section.

Reviewer

• Please provide the full name for all acronyms (such as AVEM on page 6).

Authors' reply

Done.

Reviewer

• Some questions were added to validated scales. This would be unvalidated measures unless validated elsewhere. Please specify if that was the case.

Authors' reply

We computed Cronbach’s Alpha for all measures used in the study. The psychometric properties of the DSS subscales with the additional items were better than those reported in the literature. 

Reviewer

• For ‘delay of treatment,’ which is an important outcome in this study, please provide further information on how this was assessed. There seems to be a lot of room for recall bias, which is inherently problematic and should be acknowledged in the limitations section.

Authors' reply

We agree that recall bias is a problem and we addressed this in the limitation section.

Reviewer

• The information on the LPA could potentially be included in an appendix if allowed, rather than in the body of the paper.

Authors' reply

We would like to leave the information in the methods section because the use of LPA is not so common in medical science.

Reviewer

• ‘one-way analyses of variance’= ANOVA and should be specified.

Authors' reply

In order to reduce the amount of results presented, we no longer refer to ANOVA results.

Reviewer

Results:

• Many of the covariates were not described in the methods. How is ‘higher education’ defined? College degree? More than college degree? Duration of illness is lacking units. Is this years, months, etc. Why was the cut-off of >3000 euros used? Please provide the rationale for this decision.

Authors' reply

We provided more detailed information on covariates in the statistical method section.

 The 3000 € cut-off was used because it is the average net household income in Germany at the year of the study was about 3.400 € and the mean and the median net household income category in our study were 2000-3000 € 

Reviewer

• Page 13, last sentence: the sentence structure is very convoluted, which is obscuring the meaning of the sentence. Do you mean, ‘participants in latent class 1 were less likely to have a partner…than participants in latent class 3’?

Authors' reply

We have reformulated the sentence

Reviewer

• Page 15, last paragraph, second sentence: class 2 is compared to class 2.

Authors' reply

We corrected this error

Reviewer

• The authors seem to be presenting an overwhelming amount of results that are hard to interpret. I would consider trying to figure out what the main results and take-away findings are and moving all others to an appendix section.

Authors' reply

We agree that the presented results are in part redundant. However, as a consequence of our decision to exclude participants who were treated for burnout and not for depression we also decided to present only the results for the PHQ scales and delete the results related to symptoms of burnout.

Reviewer

• Many typos throughout the results section.

Authors' reply

We did a new language editing of the whole manuscript

Reviewer

Discussion:

• The first sentence of the discussion section was not well set up in the introduction. It was unclear that this was the hypothesis that the authors were challenging.

Authors' reply

We reformulated the introduction and also the first sentence of the discussion

Reviewer

• The bullets in the discussion section are more re-hashing of the results and are actually much clearer summaries than the ones provided in the actual results section.

Authors' reply

We regard the summary of the results of the latent profile analysis as useful for helping the reader to follow the discussion.

Reviewer

• ‘Sex’ and ‘gender’ are used interchangeably throughout the paper, when the authors are in fact referring to ‘gender’ only throughout.

Authors' reply

We changed “sex” into “gender” if appropriate

Reviewer

• Page 19: there should be something in the discussion about how the info from the study could be used in practice.

Authors' reply

We discussed this point more detailed

Reviewer

Limitations section:

• Self-selection of participants is an important limitation, as it limits the generalizability of results. Can only speak for patients IN treatment who were willing to participate in the study. Cannot speak for men who do not seek treatment at all or who are unwilling to participate in the study. Generalizability should be listed as a limitation specifically.

• It is also a limitation that delay is all self-report, which is highly subjected to recall bias. Unclear how severity of illness, duration of treatment, etc may impact recall of treatment delay.

Authors' reply

We discussed these limitations in detail in the limitation section

Reviewer

• A major limitation is that type of occupation is not included in the study results. What type of work did the men included in the study do? Blue collar vs white collar? This seems like it could strongly affect or influence the findings.

Authors' reply

We included blue collar vs. white collar work as a control variable 

reviewer

Conclusions:

• Can unfortunately not speak about the ‘treatment process,’ only ‘seeking treatment’ or ‘access’ to treatment.

• The last sentence, while true, seems unrelated to the study. More could be said specifically about the impact or use of the findings.

 Authors' reply

We reformulated the conclusion to keep closer to the presented results

6. PLOS authors have the option to publish the peer review history of their article (what does this mean?). If published, this will include your full peer review and any attached files.

Do you want your identity to be public for this peer review? For information about this choice, including consent withdrawal, please see our Privacy Policy.

Reviewer #1: No

Reviewer #2: No

---

## [Decision Letter · Decision Letter 1]

1 Apr 2020

PONE-D-19-27414R1

Masculinity norms and occupational role orientation in men treated for depression.

PLOS ONE

Dear Mr. Kilian,

Thank you for submitting your manuscript to PLOS ONE. After careful consideration, we feel that it has merit but does not fully meet PLOS ONE’s publication criteria as it currently stands. Therefore, we invite you to submit a revised version of the manuscript that addresses the points raised during the review process.

We would appreciate receiving your revised manuscript by April 30, 2020. To enhance the reproducibility of your results, we recommend that if applicable you deposit your laboratory protocols in protocols.io, where a protocol can be assigned its own identifier (DOI) such that it can be cited independently in the future. For instructions see: http://journals.plos.org/plosone/s/submission-guidelines#loc-laboratory-protocols

We look forward to receiving your revised manuscript.

Kind regards,

Stephan Doering, M.D.

Academic Editor

PLOS ONE

Reviewers' comments:

Reviewer's Responses to Questions

**Comments to the Author**

1. If the authors have adequately addressed your comments raised in a previous round of review and you feel that this manuscript is now acceptable for publication, you may indicate that here to bypass the “Comments to the Author” section, enter your conflict of interest statement in the “Confidential to Editor” section, and submit your "Accept" recommendation.

Reviewer #1: All comments have been addressed

Reviewer #2: (No Response)

2. Is the manuscript technically sound, and do the data support the conclusions?

Reviewer #1: Yes

Reviewer #2: Partly

3. Has the statistical analysis been performed appropriately and rigorously? 

Reviewer #1: I Don't Know

Reviewer #2: I Don't Know

4. Have the authors made all data underlying the findings in their manuscript fully available?

Reviewer #1: Yes

Reviewer #2: Yes

5. Is the manuscript presented in an intelligible fashion and written in standard English?

Reviewer #1: Yes

Reviewer #2: No

6. Review Comments to the Author

Reviewer #1: (No Response)

Reviewer #2: Overall:

An important topic, important to help improve services to this high-risk group for suicide. Improved from last version of the manuscript. However, the manuscript still feels long, with redundancies between tables and text and could be streamlined further. There is a tendency to assume causation rather than correlation, so this should be reviewed and changed accordingly throughout the manuscript. An important contribution to the paper, but still needs revisions.

Abstract:

-there are a few typos throughout

-the conclusions should be revised to reflect the findings of the paper. While the conclusions bring up an important point, they are not directly related to the study findings. They assume causation, rather than correlation, between traditional masculinities and adequate coping strategies.

Introduction:

-pg. 4: overall the first paragraph is hard to follow. ‘Positive’ and ‘negative’ are sometimes used to refer to the direction of the association and sometimes to quality of emotions. For example, the paragraph uses ‘positive health’, ‘positive depressive symptoms’ and ‘positively related.’ Ways to steer from these words, or to use them consistently, would be useful.

-pg. 4, 2nd paragraph: ‘the question remains how this knowledge can be used to prevent or treat depressive illness in men.’ While this is true, this is not addressed in the current study. This research is not addressing that issue, and this should either be moved to the discussion or removed.

-pg 5: similarly, ‘the etiology and the course of male depression.’ This study only looks at correlations, not etiology or causation. Additionally, ‘it is currently unclear how to identify and how to deal with noxious masculinity orientations in the treatment of depression.’ While this is true, again, this will not be addressed with the current study. This is not an aim that this research can address and should be in discussion as next steps.

Methods:

-pg. 6: study design. What does ‘treatment’ for depression entail? Do you have data on men treated with meds vs therapy vs both? This could speak to severity of symptoms.

-pg 7: there is a distinction drawn between psychiatric and psychosomatic hospitals. Are these different or the same thing?

-pg. 9: what does the acronym ‘AVEM’ stand for?

-pg. 10: good clarification of duration of untreated illness from prior draft

-pg. 11, I worry that the analyses are overcontrolling by controlling for both household income and blue-collar work. Similarly, by controlling for recruitment setting, I wonder if this controlling for severity of symptoms as well. Were there major differences controlling and not controlling for those factors in the models?

-pg 12: anxiety (PHQ-7)= anxiety (GAD-7), not PHQ-7.

-pg. 12: similar worry about overadjustment with unemployment and income and blue-collar work all included in the model. Duration of mental illness should be ‘reported duration’

Results:

-pg. 13: of the individuals who responded via press invitation, where did they receive their treatment for depression?

-there are inconsistencies in rounding throughout the paper. Sometimes ‘59%’, other times 37.2%. Rounding should be consistent throughout the paper based on journal guidelines.

-pg. 13: I am not sure what ‘Abitur’ is. Please explain for an international audience.

-can psychometric properties of the study instruments in the current sample be moved to an appendix to shorten the paper which currently feels too long?

-I wonder if the class differences (pg 16) could be summarized in a table for ease of interpretation. A simple 3 x 3 table of the 3 classes, and ‘status’ ‘toughness’ and ‘anti-femininity.’

-pg. 18 and on, the RR and p-values are already listed in the table, so I would remove them in the text for clarity.

-pg. 19: ‘class 3 have higher symptoms of depression…” Should this be ‘lower’?

-there are many mistakes in the text compared to the tables. It points to a lack of attention to detail.

-there is a lot of repetition between the tables and the text. Can cut out one or the other to shorten the manuscript.

-figure 3 could be included as a supplement, rather than in the main body of the manuscript.

-can summarize the path analyses, starting in pg 21, as the findings were very similar to those presented previously.

-I would structure the findings similarly, so that if you are comparing class 1 to class 3, you then compare class 2 to class 3 (as opposed to class 3 to class 2). Similar structure will help in the interpretation of so many results.

-last 3 paragraphs of the results are repetitive and can be removed or moved to a supplement

Discussion:

-the summaries included in the discussion are good, but better belong in the results than in the discussion.

-the paragraph that begins with ‘Nevertheless, our findings’, pg. 23, should be the first paragraph of the discussion and should begin with ‘Our findings.’ The prior paragraph includes important information that seems to be more appropriate in a ‘limitations’ section. You should begin with a discussion of findings, putting them in context, then include the limitations of the current study.

-some of the discussion points are assuming causation, rather than correlation of the study findings. For example, pg. 24, 1st paragraph, ‘our data suggest that this relief may come with the price of losing crucial emotional rewards.’ Can’t draw this conclusion from the current study design, can’t assume cause and effect.

Limitations:

-memory bias, may be better termed as ‘recall bias’ which is widely used in the literature.

7. PLOS authors have the option to publish the peer review history of their article (what does this mean?). If published, this will include your full peer review and any attached files.

Reviewer #1: No

Reviewer #2: No

---

## [Author Response · Author response to Decision Letter 1]

24 Apr 2020

Reviewers' comments:

Reviewer's Responses to Questions

Comments to the Author

6. Review Comments to the Author

Reviewer #1: (No Response)

Reviewer #2: Overall:

An important topic, important to help improve services to this high-risk group for suicide. Improved from last version of the manuscript. However, the manuscript still feels long, with redundancies between tables and text and could be streamlined further. There is a tendency to assume causation rather than correlation, so this should be reviewed and changed accordingly throughout the manuscript. An important contribution to the paper, but still needs revisions.

Authors' reply

• We thank the reviewer for the general positive assessment and advice for the further improvement of the paper.

Abstract:

-there are a few typos throughout

-the conclusions should be revised to reflect the findings of the paper. While the conclusions bring up an important point, they are not directly related to the study findings. They assume causation, rather than correlation, between traditional masculinities and adequate coping strategies.

Authors' reply

• We apologize for the typos and corrected them throughout the text.

• We revised the conclusion section of the abstract as advised 

Introduction:

-pg. 4: overall the first paragraph is hard to follow. ‘Positive’ and ‘negative’ are sometimes used to refer to the direction of the association and sometimes to quality of emotions. For example, the paragraph uses ‘positive health’, ‘positive depressive symptoms’ and ‘positively related.’ Ways to steer from these words, or to use them consistently, would be useful.

-pg. 4, 2nd paragraph: ‘the question remains how this knowledge can be used to prevent or treat depressive illness in men.’ While this is true, this is not addressed in the current study. This research is not addressing that issue, and this should either be moved to the discussion or removed.

-pg 5: similarly, ‘the etiology and the course of male depression.’ This study only looks at correlations, not etiology or causation. Additionally, ‘it is currently unclear how to identify and how to deal with noxious masculinity orientations in the treatment of depression.’ While this is true, again, this will not be addressed with the current study. This is not an aim that this research can address and should be in discussion as next steps.

Authors' reply

• We thank the reviewer for these comments and revised the paragraphs as advised.

Methods:

-pg. 6: study design. What does ‘treatment’ for depression entail? Do you have data on men treated with meds vs therapy vs both? This could speak to severity of symptoms.

Authors' reply

• We did not ask our participants after the type of treatment they received. However, we assessed the severity of depressive symptoms by means of the PHQ-9 cut-off scores which are provided in Table 1.

• We added the information in the method section, that we included patients treated with medications or with psychotherapy or both. 

-pg 7: there is a distinction drawn between psychiatric and psychosomatic hospitals. Are these different or the same thing?

Authors' reply

• We tried to improve our explanation of the differences between psychiatric and psychosomatic hospitals by the following formulation at pg 7. 

-pg. 9: what does the acronym ‘AVEM’ stand for?

Authors' reply

• We added the German meaning of the acronym at page 9.

-pg. 10: good clarification of duration of untreated illness from prior draft

Authors' reply

Thanks.

-pg. 11, I worry that the analyses are overcontrolling by controlling for both household income and blue-collar work. Similarly, by controlling for recruitment setting, I wonder if this controlling for severity of symptoms as well. Were there major differences controlling and not controlling for those factors in the models?

Authors' reply

• We thank the reviewer for this important comment. We checked for multicollinearity by estimating the variance inflation factor (VIF) for each model variable and the average VIF for each regression model, revealing no indication for multicollinearity. In addition, we compared the adjusted results with the unadjusted results, indicating that the regression coefficients and the estimated means are quite similar and that the interpretation of the results remains the same after adjustment. Nevertheless, we think that adjustment for the socioeconomic variables is reasonable because it provides the effects of masculinity and job related attitudes independent of the confounding effects of economic conditions, education and the work environment. With regard to the inconclusive results we found in the literature, this seems important.

-pg 12: anxiety (PHQ-7)= anxiety (GAD-7), not PHQ-7.

Authors' reply

• We apologize for these mistakes and corrected them.

-pg. 12: similar worry about overadjustment with unemployment and income and blue-collar work all included in the model. Duration of mental illness should be ‘reported duration’

Authors' reply

• Regarding overadjustment, see above.

• We changed DUI into “reported DUI”.

Results:

-pg. 13: of the individuals who responded via press invitation, where did they receive their treatment for depression?

Authors' reply

• We provided this information at pg 14.

-there are inconsistencies in rounding throughout the paper. Sometimes ‘59%’, other times 37.2%. Rounding should be consistent throughout the paper based on journal guidelines.

Authors' reply

• We corrected the inconsistencies.

-pg. 13: I am not sure what ‘Abitur’ is. Please explain for an international audience.

Authors' reply

• We explained the term in the method section at p. 11.

-can psychometric properties of the study instruments in the current sample be moved to an appendix to shorten the paper which currently feels too long?

Authors' reply

• We moved this part of the results into the appendix.

-I wonder if the class differences (pg 16) could be summarized in a table for ease of interpretation. A simple 3 x 3 table of the 3 classes, and ‘status’ ‘toughness’ and ‘anti-femininity.’

Authors' reply

• Adding a table with the class differences would lead to a doubble display of the information already presented in figure 1. However, we followed this recommendation and added a new table S2, providing the means, the standard deviations and the results of the ANOVAs for the dimensions of the masculinity and the job-related attitudes to the online supplement.

-pg. 18 and on, the RR and p-values are already listed in the table, so I would remove them in the text for clarity.

Authors' reply

• We followed this recommendation and removed the RRs from the text.

-pg. 19: ‘class 3 have higher symptoms of depression…” Should this be ‘lower’?

-there are many mistakes in the text compared to the tables. It points to a lack of attention to detail.

Authors' reply

• We corrected these mistakes.

-there is a lot of repetition between the tables and the text. Can cut out one or the other to shorten the manuscript.

-figure 3 could be included as a supplement, rather than in the main body of the manuscript.

Authors' reply

• We followed the advice and moved Table 4 and figure 3 into the supplement.

-can summarize the path analyses, starting in pg 21, as the findings were very similar to those presented previously.

-I would structure the findings similarly, so that if you are comparing class 1 to class 3, you then compare class 2 to class 3 (as opposed to class 3 to class 2). Similar structure will help in the interpretation of so many results.

Authors' reply

• We followed the advice and changed the order of the comparison.

-last 3 paragraphs of the results are repetitive and can be removed or moved to a supplement

Authors' reply

• The last three paragraphs provide information about indirect effects and about the model fit of the path model. This information is needed for the interpretation of the path analysis. Therefore we want to leave this information in the main text. However, we combined the last two paragraphs into one.

Discussion:

-the summaries included in the discussion are good, but better belong in the results than in the discussion.

Authors' reply

• We followed the advice and moved the summary of the LPA results into the results section.

-the paragraph that begins with ‘Nevertheless, our findings’, pg. 23, should be the first paragraph of the discussion and should begin with ‘Our findings.’ The prior paragraph includes important information that seems to be more appropriate in a ‘limitations’ section. You should begin with a discussion of findings, putting them in context, then include the limitations of the current study.

Authors' reply

• We followed the advices and restructured the discussion.

-some of the discussion points are assuming causation, rather than correlation of the study findings. For example, pg. 24, 1st paragraph, ‘our data suggest that this relief may come with the price of losing crucial emotional rewards.’ Can’t draw this conclusion from the current study design, can’t assume cause and effect.

Authors' reply

• We thank the reviewer for this important advice and reformulated this conclusion into a hypothesis to be investigated in further studies.

Limitations:

-memory bias, may be better termed as ‘recall bias’ which is widely used in the literature.

Authors' reply

• We gladly follow this advice.

---

## [Decision Letter · Decision Letter 2]

13 May 2020

Masculinity norms and occupational role orientations in men treated for depression.

PONE-D-19-27414R2

Dear Dr. Kilian,

We are pleased to inform you that your manuscript has been judged scientifically suitable for publication and will be formally accepted for publication once it complies with all outstanding technical requirements.

With kind regards,

Stephan Doering, M.D.

Academic Editor

PLOS ONE

Reviewer's Responses to Questions

**Comments to the Author**

1. If the authors have adequately addressed your comments raised in a previous round of review and you feel that this manuscript is now acceptable for publication, you may indicate that here to bypass the “Comments to the Author” section, enter your conflict of interest statement in the “Confidential to Editor” section, and submit your "Accept" recommendation.

Reviewer #3: All comments have been addressed

2. Is the manuscript technically sound, and do the data support the conclusions?

Reviewer #3: Yes

3. Has the statistical analysis been performed appropriately and rigorously? 

Reviewer #3: Yes

4. Have the authors made all data underlying the findings in their manuscript fully available?

Reviewer #3: Yes

5. Is the manuscript presented in an intelligible fashion and written in standard English?

Reviewer #3: Yes

6. Review Comments to the Author

Reviewer #3: The criticism of the previous reviewers has been addressed comprehensively and sufficiently. The manuscript can be accepted as it is.

7. PLOS authors have the option to publish the peer review history of their article (what does this mean?). If published, this will include your full peer review and any attached files.

Reviewer #3: No

---

## [Editor Report · Acceptance letter]

15 May 2020

PONE-D-19-27414R2 

Masculinity norms and occupational role orientations in men treated for depression. 

Dear Dr. Kilian:

I am pleased to inform you that your manuscript has been deemed suitable for publication in PLOS ONE. Congratulations! Your manuscript is now with our production department. 

With kind regards,

on behalf of

Professor Stephan Doering 

Academic Editor

PLOS ONE